# Clinical and Prognostic Impact of Hemodynamic Gain Index and Heart Hemodynamic Reserve in Heart Failure with Reduced and Mildly Reduced Ejection Fraction: A Multicenter Study

**DOI:** 10.3390/diagnostics15182366

**Published:** 2025-09-17

**Authors:** Emiliano Fiori, Sara Corradetti, Giovanna Gallo, Alberto Palazzuoli, Antonio Pagliaro, Roberta Molle, Pier Giorgio Tiberi, Elisabetta Salvioni, Arianna Piotti, Paola Gugliandolo, Piergiuseppe Agostoni, Damiano Magrì, Emanuele Barbato

**Affiliations:** 1Department of Clinical and Molecular Medicine, Sapienza University, 00189 Rome, Italy; 2Cardiovascular Center Aalst, AZORG-Clinic, 9300 Aalst, Belgium; 3Cardiovascular Diseases Unit, Cardio-Thoracic and Vascular Department, S. Maria alle Scotte Hospital, University of Siena, 53100 Siena, Italy; 4Centro Cardiologico Monzino, IRCCS, 20138 Milan, Italy; 5Department of Clinical sciences and Community Health, Cardiovascular Section, University of Milano, 20122 Milan, Italy

**Keywords:** heart failure, cardiopulmonary exercise test, hemodynamic gain index, chronotropic incompetence

## Abstract

**Background/Objectives:** Cardiopulmonary exercise testing (CPET) is a well-established tool for risk stratification in patients with heart failure (HF); however, its utility is limited in routine clinical practice due to the associated cost and technical demands. The hemodynamic gain index (HGI), a non-metabolic parameter derived from systolic blood pressure and heart rate changes during exercise, has been demonstrated to play a promising role in HF populations. In this study, we aimed both to validate the prognostic value of the HGI and to evaluate a novel metric, heart hemodynamic reserve (HHR), in patients with HF and left ventricular ejection fraction (LVEF) below 50%. **Methods:** We retrospectively enrolled 479 consecutive patients with HF and reduced or mildly reduced LVEF who underwent maximal, symptom-limited CPET at three Italian university hospitals between 2012 and 2024. The HGI and HHR were computed using resting and peak exercise hemodynamic data. HHR is defined as the product of systolic blood pressure and heart rate reserve with exercise, normalized for the age-predicted maximum heart rate. The primary endpoint was a composite of cardiovascular death, urgent heart transplantation (HTx), or left ventricular assist device (LVAD) implantation. Prognostic associations were assessed using multivariable Cox regression and area under the receiver operating characteristic curves (AUCs). **Results:** During a median follow-up of 3.25 years, the composite outcome occurred in 56 patients (11.5%). Both the HGI and HHR were independently associated with the prespecified endpoint (HGI HR: 0.41, 95% CI: 0.20–0.83, *p* = 0.013; HHR HR: 0.89, 95% CI: 0.83–0.96, *p* = 0.004), with HHR showing a slightly higher prognostic accuracy than the HGI (AUC 0.78 vs. 0.74; *p* = 0.033). **Conclusions:** Both the HGI and HHR are independent prognostic markers in HF patients with LVEF < 50%. Their non-metabolic derivation makes them valuable tools for risk stratification in settings where CPET is unavailable.

## 1. Introduction

Heart failure syndrome (HF) is a global epidemic that places a significant burden on healthcare systems worldwide. Despite continuous advancements in the field of HF medications, HF with reduced ejection fraction (HFrEF) remains the major cause of hospital admission in adults and is associated with a high risk of cardiovascular mortality. HF with mildly reduced ejection fraction (HFmrEF) is considered a milder form of HFrEF, with a more favorable prognostic profile albeit a lesser studied pathophysiology [1].

Functional assessment with a cardiopulmonary exercise test (CPET) is considered a strategic tool for the fine prognostic assessment of HF, with all its facets of ejection fraction [2,3,4]. However, CPET is both resource-intensive and time-consuming, which limits its practicality in many healthcare environments. The six-minute walk test (6MWT) is a possible and more widely available alternative to CPET, and its value in monitoring HF patients has recently been demonstrated [5,6,7]. Nevertheless, the absence of strong standardization and the test’s poor reproducibility hamper the reliability of using the 6MWT for prognostic assessments [8]. Therefore, current research is focusing on evaluating novel non-metabolic exercise testing-derived parameters, which could represent solid alternatives to classical CPET variables in the prognostic assessment of HF [9] in resource-limited settings. In such a context, the hemodynamic gain index (HGI) might be a promising candidate [10], reflecting the relative increase in the peak rate pressure product (RPP). The HGI has been studied and has prognostic relevance in different phenotypes of HF patients, including preserved EF (HFpEF) [11], and has been evaluated using both bicycle [9] and treadmill exercise tests [12]. However, the existing evidence is primarily derived from single-center studies, and its role in the “gray zone” of HFmrEF has yet to be defined. Moreover, as demonstrated by Chaikijurajai et al., the HGI is highly dependent on the chronotropic response during exercise [9], which may limit its accuracy in reflecting true heart hemodynamic reserve, especially with advancing age and in the presence of chronotropic incompetence, both of which are common conditions among HF patients [13]. 

This study aims to validate and support the current understanding of the HGI by extending the evaluation of its prognostic relevance to patients with HFmrEF. We further hypothesized that a similar non-metabolic exercise-test-derived parameter incorporating relative heart rate reserve may offer comparable, if not superior, prognostic accuracy in this population.

## 2. Methods

### 2.1. Study Population

We retrospectively analyzed 970 consecutive patients with known HF and LVEF < 50% who were referred for CPET at three university centers in Italy (Azienda Ospedaliero Universitaria Sant’Andrea, Rome; Azienda Ospedaliera Universitaria Senese, Siena; and Centro Cardiologico Monzino, IRCCS, Milan, Italy) between January 2012 and January 2024.

Eligible patients were required to be in stable clinical condition, with unchanged medications for at least three months, and to have undergone a maximal symptom-limited CPET on a cycle ergometer.

Patients were excluded if they had comorbidities directly interfering with exercise performance, such as moderate-to-severe anemia (hemoglobin < 10 g/dL), severe obstructive or restrictive lung disease, significant peripheral vascular disease, or exercise-induced angina and/or ST-segment changes. Patients with severe primary mitral or aortic valve disease, atrial fibrillation, or paced rhythm due to atrioventricular nodal dysfunction, as well as those with missing vital parameters at baseline or peak exercise, were also excluded from the present analysis.

Baseline clinical data were obtained from the institutional registries, and standard echocardiographic assessments were performed according to current recommendations.

### 2.2. Cardiopulmonary Exercise Test: HGI and HHR

A maximal, symptom-limited CPET was performed on an electronically braked cycle ergometer connected to a metabolic chart. A personalized ramp exercise protocol was chosen, aiming at a test duration of 10 ± 2 min. The exercise was preceded by a standard technique, specifically by a 2-min resting phase with breath-by-breath gas exchange monitoring followed by a 3-min unloaded warm-up. CPET was self-terminated by the patient upon reporting maximal effort and as confirmed by a peak respiratory exchange ratio (RER) ≥ 1.05. A breath-by-breath analysis of O_2_, carbon dioxide (CO_2_), and ventilation (VE) was performed, and peak values were computed as the highest observed measurements (20 s average). The predicted pVO_2_ was determined by using the sex, age and weight-adjusted Hansen/Wasserman equations. The anaerobic threshold (AT) was identified through a V-slope analysis of VO_2_ and CO_2_ production (VCO_2_) and was confirmed through the specific behavior of the ventilatory equivalents of O_2_ (VE/VO_2_) and CO_2_ (VE/VCO_2_) [14]. The relation between VE and VCO_2_ (i.e., the ventilatory efficiency) was analyzed as the slope (VE/VCO_2_ slope) of the linear relationship between VE and VCO_2_ from 1 min after the beginning of the loaded exercise to the end of the isocapnic buffering period [15]. A 12-lead ECG, blood pressure, and HR were recorded. Specifically, peak HR (pHR) and peak systolic blood pressure (pSBP) were collected during CPETs, whereas baseline HR (bHR) and SBP (bSBP) were measured before the test after at least 2 min of rest in a seated position on the cycle ergometer. Circulatory power was calculated from the following formula: pSBP × peak VO_2_/Kg. pHR and bHR were also analyzed as a percentage of the maximum predicted value according to the standard formula: [16,17,18] pHR% = [(pHR/220 − age) × 100]; bHR% = [(bHR/220 − age) × 100]. The HGI was calculated using the following formula: [(pSBP × pHR) − (bSBP × bHR)]/(bSBP × bHR). The heart hemodynamic reserve (HHR) was developed using the change in HR (as percentage of maximum predicted) and SBP from baseline to peak exercise. The following equation was constructed: HHR = [(pHR% − bHR%) × (pSBP − bSBP)].

### 2.3. Study Objectives

The objective of this study was to assess the prognostic implications of the HGI and HHR in a contemporary multicenter-derived population of HFrEF and HFmrEF patients. The primary outcome was a composite of cardiovascular (CV) death, left ventricular assist device (LVAD) implantation, and urgent heart transplantation (HTX) over time. The secondary aim of this study was to compare the HGI and HHR across different subgroups.

### 2.4. Statistical Analysis

For baseline characteristics, the normal distribution of continuous variables was assessed using the Shapiro–Wilk test. Continuous variables were expressed as mean ± standard deviation or median (interquartile range). Categorical variables were expressed as counts and percentages. Differences between groups were analyzed using the *t*-test or the Mann–Whitney U-test for continuous variables and the chi-square test or the Fisher’s exact test for categorical variables, as appropriate.

Hazard ratios (HRs) for primary composite outcome and corresponding 95% CIs per increase in the HGI and HHR and between tertiles of the HGI and HHR were estimated using both the univariable and multivariable Cox proportional hazards models adjusted for beta-blocker (BB) treatment, chronic diuretic therapy, LVEF, ischemic etiology, hemoglobin (Hb), pVO_2_%, and VE/VCO_2_ slope. The variables included in the multivariable model were selected from those that showed a significant association with the outcome in the univariate analysis. Kaplan–Meier plots truncated at two years along with the log-rank test were used to compare the estimated cumulative rates of survival free from the composite outcome. We also assessed the prognostic significance of the HGI and HHR with subgroup analysis on the basis of the following variables: age (≥65 or <65 years), sex, etiology (ischemic or non-ischemic), beta-blocker use, exercise effort (maximal or submaximal, determined by an RER cutoff of 1.05), and significant chronotropic incompetence (defined as pHR% < 65 [19,20]).

In addition, receiver operating characteristic (ROC) curves with corresponding areas under the curve (AUCs) were estimated to demonstrate the discriminative capability of the HGI, HHR, pVO_2_/kg, percent predicted pVO_2_ (pVO_2_%), percent peak heart rate (pHR%), circulatory power, and VE/VCO_2_ slope. DeLong’s test was used to compare AUCs. Optimal cutoff values for HGI and HHR were determined using the Youden index.

All analyses were performed using R software version 4.5.0 (R Foundation, Vienna, Austria), and values of *p* < 0.05 were considered statistically significant.

## 3. Results

### 3.1. General Characteristics of the Study Population

A total of 479 patients were ultimately included in the analyses: 393 with HFrEF and 86 with HFmrEF. The demographic and clinical characteristics of the entire study sample are reported in Table 1. The study population mainly consisted of middle-aged male patients in NYHA functional class II and III. HF treatment was in accordance with the guidelines [21], with each class considered as optimized by the HF cardiologist in charge of the patient. At study entry, an implantable cardioverter-defibrillator was present in nearly 40% of the entire population, alone or in combination with biventricular pacing (Table 1).

The exercise and CPET variables depict a population with moderate-to-severe exercise limitation (mean pVO_2_% 60 ± 20), reduced ventilatory efficiency (VE/VCO_2_ slope 34.9 ± 9.1), and impaired chronotropic competence (mean pHR% 0.70 ± 0.1). The mean HGI was 1.2 ± 0.7, while the mean HHR was 9.7 ± 9.5.

During a median follow-up of 3.25 years, the composite outcome of CV death, LVAD implantation, or HTX occurred in 56 patients (11.5%). Among all the HFmrEF patients, one was listed and transplanted due to recurrent ventricular arrhythmias. Among all the HFrEF patients, there were 30 CV deaths (7.8%) and 25 LVAD implantations/HTX (6.3%). Three patients with HFrEF died of non-CV causes (one sepsis and two cerebrovascular accident).

### 3.2. Functional Characterization of HGI and HHR

When stratifying the population into tertiles based on the HGI (T1: HGI < 0.87; T2: 0.87 ≤ HGI <1.43; T3: HGI > 1.43) and on HHR (T1: HHR < 3.97; T2: 3.97 ≤ HHR < 10.7; T3: HHR > 10.7), a similar pattern of CPET variables across the tertiles was observed (Figure 1). Specifically, reductions in the HGI and HHR were paralleled by a reduction in functional capacity (pVO_2_%) and CP, while lower HGI and HHR values were associated with worse ventilatory efficiency and chronotropic competence (Figure 1, see Appendix A).

### 3.3. Prognostic Relevance of HGI and HHR

Table 2 reports the detailed univariate analysis of the main variables. Among the CPET variables, pHR%, pVO_2_/Kg, pVO_2_%, and CP showed a negative association with the composite outcome, while the VE/VCO2 slope was positively associated (HR per 1-unit increase in VE/VCO_2_ slope: 1.06, CI 1.04–1.08, *p* < 0.001). Both the HGI and HHR demonstrated a negative protective association with the composite outcome, with increased values of the two indexes being associated with a lower risk of events (HR per 1-unit increase in HGI: 0.22, CI 0.12–0.39, *p* < 0.001; HR per 1-unit increase in HHR: 0.85, CI 0.79–0.90, *p* < 0.001).

When treated as continuous variables, the HGI remained the only exercise-related factor significantly associated with the composite outcome after multivariable regression adjusting for LVEF, ischemic etiology, Hb, beta-blocker and diuretic treatment, pVO_2_%, and VE/VCO_2_ slope (adjusted HR per 1-unit increase in HGI: 0.44, CI 0.23–0.83, *p* = 0.011) (Table 3). Subgroup analysis consistently showed that a higher HGI was associated with a lower risk of the composite outcome, regardless of age, sex, HF etiology, stable treatment with beta-blocker, exercise test maximality, or chronotropic incompetence (all *p* for interaction > 0.05) (Figure 2).

We therefore stratified the population into tertiles based on the HGI (T1: HGI < 0.87; T2: 0.87 ≤ HGI < 1.43; T3: HGI > 1.43). Kaplan–Meier survival curves showed that patients in the lower HGI tertiles (T1 and T2) had a significantly higher incidence of the composite outcome at two years (log-rank *p* < 0.001) (Figure 3). In Cox regression with tertiles, the highest tertile (T3) was used as the reference group (HR = 1). The lowest HGI values (T1) were significantly associated with an increased risk of the composite outcome in both Cox univariable analyses (T1 vs. T3: HR 12.77, 95% CI 3.94–41.45, *p* < 0.001) and multivariable adjustment (T1 vs. T3: HR 5.35, CI 1.57–18.22, *p* = 0.007) (Table 4). Despite not reaching statistical significance after multivariable adjustment, patients in the intermediate tertile (T2) showed a clear trend towards a higher risk of events compared to T3 (T2 vs. T3: HR 5.76, CI 1.68–19.77, *p* = 0.005; adjusted HR: 3.47, CI 0.98–12.24, *p* = 0.053). The findings were therefore consistent whether the HGI was treated as a continuous (Table 2 and Table 3) or categorical variable (Table 4).

As with the HGI, HHR as a continuous variable remained significantly associated with the composite outcome after multivariable regression (adjusted HR per 1-unit increase in HHR: 0.91, CI 0.85–0.97, *p* = 0.005) (Table 3). Subgroup analysis revealed a similar pattern for HHR as observed with the HGI, with consistent associations across all subgroups (Figure 2).

We therefore stratified the population into tertiles based on HHR (T1: HHR < 3.97; T2: 3.97 ≤ HHR < 10.7; T3: HHR > 10.7). Kaplan–Meier curves showed that only patients in the lowest HHR tertile (T1) had a significantly higher incidence of the composite outcome at two years (log-rank *p* < 0.001) (Figure 3). In Cox regression, only the lowest tertile (T1) of HHR was significantly associated with a higher risk of the composite outcome (T1 vs. T3: HR 9.03, 95% CI 3.56–22.9, *p* < 0.001). This association persisted after multivariable adjustment (T1 vs. T3: HR 3.75, CI 1.37–10.25, *p* = 0.010) (Table 4).

### 3.4. Prognostic Accuracy of HGI and HHR

To assess the prognostic performance of the HGI in predicting the composite outcome of CV death, HTx, and LVAD implantation, we compared its discriminative capacity with that of traditional CPET-derived variables. The HGI demonstrated an AUC of 0.74, which was comparable to that of CP (AUC = 0.77; *p* = 0.32), VE/VCO_2_ slope (AUC = 0.76; *p* = 0.66), pVO_2_% (AUC = 0.73; *p* = 0.66), and pVO_2_/kg (AUC = 0.69; *p* = 0.11), while outperforming pHR% (AUC = 0.66; *p* = 0.01) (Figure 4).

When comparing the HGI with HHR, which integrates relative heart rate reserve and exercise-induced systolic pressure response, the latter demonstrated significantly superior discriminative performance (AUC = 0.78 vs. 0.74; *p* = 0.033) (Figure 4). These findings underscore the potential value of both the HGI and HHR as clinically relevant tools for prognostic assessment in patients with heart failure and LVEF < 50%, with HHR potentially offering incremental prognostic insight.

An HGI cutoff of 0.88 and an HHR cutoff of 4.33 yielded the highest Youden indices, with sensitivity/specificity of 0.68/0.70 and 0.77/0.70, respectively. In the univariable Cox regression, a low HGI (<0.88) was associated with a 4.17-fold increased risk of the composite outcome (HR: 4.17; 95% CI: 2.38–7.31; *p* < 0.001), while low HHR (<4.33) conferred an even higher risk (HR: 6.25; 95% CI: 3.36–11.64; *p* < 0.001).

In the multivariable analysis adjusted for VE/VCO_2_ slope, pVO_2_%, pHR%, LVEF, and beta-blocker use, both the HGI and HHR remained independent predictors. Notably, HHR < 4.33 was associated with a 3.65-fold increased risk (adjusted HR: 3.65; 95% CI: 1.74–7.65; *p* < 0.001), while an HGI <0.88 remained significant with an adjusted HR of 2.28 (95% CI: 1.16–4.48; *p* = 0.016).

## 4. Discussion

In this multicenter retrospective cohort of patients with heart failure and LVEF below 50%, the hemodynamic gain index (HGI) was validated as an independent predictor of the HF-related primary composite outcome of CV death, LVAD implantation, or urgent HTx. The newly proposed heart hemodynamic reserve (HHR) showed significant correlations with established CPET-derived measures of functional capacity and demonstrated superior overall prognostic accuracy compared with the HGI.

Exercise-derived variables offer a valuable additive perspective in the risk stratification across the full spectrum of the HF ejection fraction [22]. In this context, CPET stands out due to its unique ability to capture the intricate interplay between cardiovascular and respiratory systems. CPET provides one of the most accurate characterizations of functional limitation and prognostication in HF patients [23]. However, despite these well-established benefits, CPET remains underutilized in clinical practice, largely because of the need for specialized expertise, as well as the financial and logistical burdens associated with the metabolic cart and its ongoing maintenance [24].

In response to these limitations, current research has increasingly focused on the revaluation and repurposing of non-metabolic, exercise-derived parameters that may retain prognostic value. These parameters can be assessed using more widely available tools such as the cycle ergometer, making them potentially more accessible in broader clinical contexts. They variably combine the blood pressure and HR response to increasing workloads. Among these metrics, the rate pressure product (RPP) and hemodynamic gain index (HGI), though closely related, serve complementary roles: the RPP reflects the maximal circulatory response achieved during exercise, while the HGI represents the available circulatory reserve mobilized in response to exertion.

The HGI, a relatively novel parameter, has recently been studied in both HFrEF [9] and HFpEF [11]. It has shown significant correlations with well-established CPET metrics of prognostic importance, such as pVO_2_ and the VE/VCO_2_ slope [9,12]. These findings suggest that the HGI may offer reliable prognostic insight into HF outcomes. Nonetheless, the existing evidence stems largely from single-center studies, and the relevance of the HGI in patients with HFmrEF remains unexplored. Notably, the present study included nearly a quarter of patients with HFmrEF, thereby addressing an important gap in the current literature. Moreover, both the RPP and HGI are heavily influenced by exercise-induced changes in heart rate [9]. This is particularly relevant in the context of HF, where SBP responses tend to be blunted and chronotropic incompetence is common, regardless of the HF phenotype [13]. Furthermore, HR dynamics during exercise can be affected by a range of factors, including beta-blocker therapy and comorbid conditions such as atrial fibrillation. To overcome these limitations and further refine exercise-based prognostication, we developed and tested a novel parameter, the heart hemodynamic reserve (HHR). This metric was specifically designed to adjust the observed increase in HR for the maximum age-predicted HR, thus normalizing the cardiovascular response relative to physiological expectations.

The present study confirms that a reduction in the HGI is associated with the deterioration of key CPET-derived metabolic parameters, including pVO_2_%, VE/VCO_2_ slope, and CP (Figure 1). In our multicenter retrospective cohort of HF patients with LVEF < 50%, the HGI exhibited a robust and consistent protective association with the primary composite outcome of CV death, HTx, or LVAD implantation. Notably, this relationship remained significant after adjustment for established exercise prognostic variables (adjusted HR: 0.41, 95% CI: 0.20–0.83; *p* = 0.013). The prognostic value of the HGI was consistent across the spectrum of reduced LVEF and was independent of test maximality, beta-blocker therapy, or the presence of severe chronotropic incompetence (Figure 2). Furthermore, the HGI demonstrated prognostic accuracy for the primary outcome that was comparable to commonly used metabolic variables (Figure 4). The optimal cutoff identified in our cohort (HGI < 0.88) was slightly higher than the threshold proposed by Chaikijurajai et al. (HGI < 0.77), though the diagnostic performance remained similar, with comparable sensitivity and specificity (0.68/0.70 vs. 0.59/0.72).

A parallel analysis was conducted for HHR. Since SBP increase during exercise cannot be reliably predicted, HHR was designed to account for individual chronotropic reserve by adjusting both resting and peak exercise heart rates relative to the maximum age-predicted heart rate. In essence, HHR reflects the product of the SBP increase (pSBP − bSBP) and the relative increase in HR expressed as a percentage of the predicted maximum (pHR% − bHR%).

Compared to the HGI, HHR exhibited a similar pattern: lower HHR values were associated with greater impairment in cardiopulmonary performance (Figure 1). Moreover, HHR demonstrated a comparable association with the primary composite outcome, both in univariate and multivariate analyses (adjusted HR: 0.89, 95% CI: 0.83–0.96; *p* = 0.004), and this association remained consistent across all prespecified subgroups (Figure 2). While the HGI seems more broadly predictive across both lower tertiles, HHR’s prognostic value seems concentrated in patients with the lowest HHR values (i.e., T1) (Figure 3). In terms of prognostic performance, HHR provided superior accuracy compared to the HGI (Figure 4).

While both the HGI and HHR demonstrated independent prognostic value, their slightly different behavior may be explained by their respective physiological underpinnings. The HGI, which is based on absolute changes in heart rate and systolic blood pressure during exercise, appears more sensitive across a wider spectrum of functional impairment, effectively stratifying risk in both the lowest and intermediate tertiles. In contrast, HHR incorporates age-predicted maximum heart rate to normalize chronotropic response, thereby emphasizing physiologic reserve relative to individual expectations. This normalization likely enhances the specificity of HHR for identifying patients with true circulatory limitations, particularly those with severe chronotropic incompetence or impaired systolic blood pressure response. Consequently, although the HGI stratifies risk more broadly, HHR demonstrated superior overall prognostic accuracy, as reflected by a higher area under the ROC curve (AUC: 0.78 vs. 0.74) (Figure 4). These findings suggest that while the HGI may offer broader clinical applicability across diverse risk profiles, HHR provides a more precise measure of physiologic reserve and may be particularly useful in identifying patients at highest risk.

## 5. Study Limitations

Our study has several limitations. First, patients with atrial fibrillation [25,26] or paced rhythms [27] were excluded to allow accurate evaluation of heart rate kinetics. While methodologically necessary, this may have introduced a selection bias toward a clinically more stable population. This is also reflected in the relatively low event rate (11.5%) observed over a median follow-up of 3.25 years. Although the study included a substantial proportion of patients with HFmrEF, the very low-risk profile of this phenotype [28], evidenced by only one observed event, statistically precluded a reliable subgroup analysis in this group. Second, the retrospective design carries an inherent risk of unmeasured confounding, despite multivariable adjustments.

Although beta-blocker use was accounted for, the potential modifying effects of other HF therapies were not fully explored. In particular, the apparently harmful association observed for SGLT2 inhibitors in the univariate analysis should be interpreted with caution. This finding is likely attributable to confounding by indication, as SGLT2i prescription was strongly influenced by the enrollment period and the heterogeneity of the study population. Patients with HFmrEF who had limited indications, lower drug availability, and a more favorable clinical profile rarely received SGLT2i and showed a very low incidence of events. With the exception of beta-blockers, which may influence heart rate dynamics and thereby affect the parameters under study, and chronic diuretic therapy, which has been consistently available in routine HF care and represents a well-established marker of advanced disease, all other medications were excluded from the multivariate model.

The underrepresentation of women in our cohort further limits the generalizability of the findings, as sex-specific differences in hemodynamic and exercise responses may be clinically relevant [29,30]. Finally, HHR characterization and cutoff values were derived post hoc and validated within the same cohort, which may have led to an overestimation of prognostic accuracy and highlights the need for validation in independent populations.

## 6. Conclusions

The HGI and the newly proposed HHR emerge as robust, independent predictors of adverse outcomes in patients with heart failure and LVEF below 50%. Importantly, HHR demonstrated superior overall prognostic accuracy compared with the HGI, likely owing to its ability to normalize for age-predicted chronotropic reserve.

Given their strong associations with cardiopulmonary performance and clinical outcomes, the HGI and HHR provide complementary insights into circulatory reserve and functional limitations. Their derivation from standard non-metabolic exercise testing parameters makes them particularly attractive in clinical settings where access to CPET is limited. These findings support the integration of the HGI into routine exercise assessments to improve the risk stratification of HF patients across the spectrum of EF.

## Figures and Tables

**Figure 1 diagnostics-15-02366-f001:**
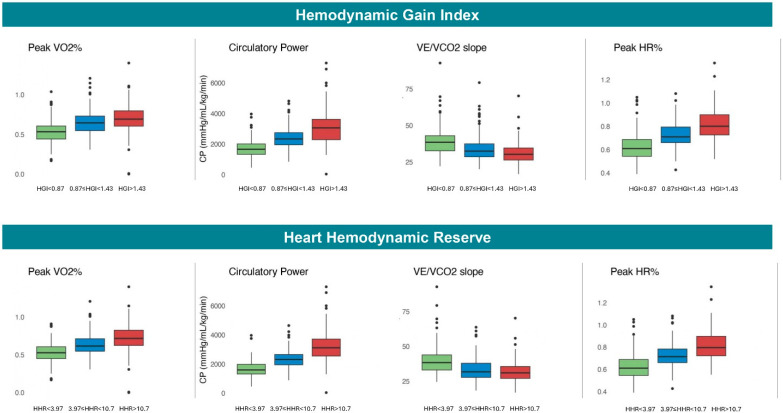
Behavior of CPET parameters according to HGI and HHR tertiles. Both HGI and HHR show a consistent relationship with established exercise-derived variables. Lower values of HGI and HHR (T1 vs. T3) are associated with reduced functional capacity (pVO_2_%), lower circulatory power (CP), and impaired ventilatory efficiency (higher VE/VCO_2_ slope), well-recognized predictors of adverse prognosis in HF. **Abbreviations**: HGI, hemodynamic gain index; HHR, heart hemodynamic reserve; pVO_2_%, peak oxygen consumption as percentage of maximum predicted; VE/VCO_2_ slope, relationship between ventilation and carbon dioxide production.

**Figure 2 diagnostics-15-02366-f002:**
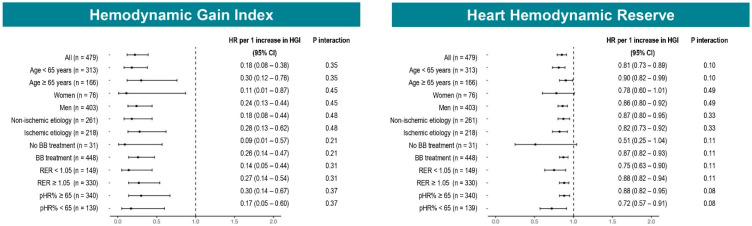
Subgroup analysis of HGI, HHR, and adverse clinical outcomes. Forest plots display the hazard ratio (HR) per 1-unit increase in HGI and HHR with 95% confidence intervals across predefined subgroups, along with *p* values for interaction. The prognostic association of HGI and HHR remained consistent across all subgroups. **Abbreviations**: BB, beta-blocker; RER, peak respiratory exchange ratio; pHR%, peak heart rate expressed as percentage of maximum predicted.

**Figure 3 diagnostics-15-02366-f003:**
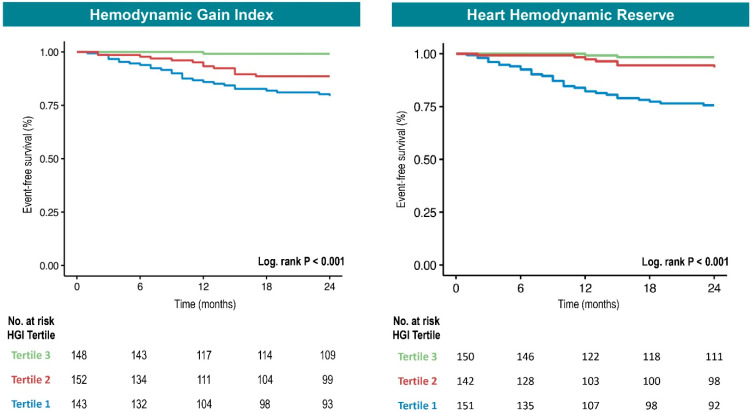
Kaplan–Meier curves for adverse clinical outcomes stratified by HGI and HHR tertiles. Event-free survival in patients with HF and LVEF < 50% according to tertiles of the hemodynamic gain index (HGI, **left**) and heart hemodynamic reserve (HHR, **right**). Patients in the lower HGI tertiles (T1 and T2 vs. T3) showed significantly reduced event-free survival for the primary composite outcome (CV death, LVAD implantation, and heart transplantation). For HHR, only the lowest tertile (T1 vs. T3) was associated with worse prognosis (log-rank *p* < 0.001). **Abbreviations**: CV, cardiovascular; HGI, hemodynamic gain index; HHR, heart hemodynamic reserve; LVEF, left ventricular ejection fraction; LVAD, left ventricular assist device.

**Figure 4 diagnostics-15-02366-f004:**
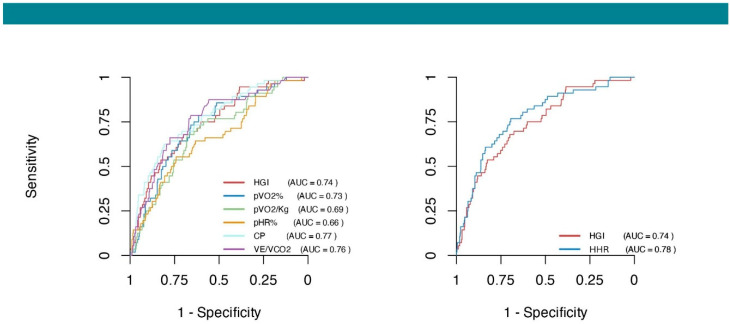
Receiver operating characteristic curves for the HGI, HHR, and other CPET parameters. ROC curves illustrating the discriminative performance of the hemodynamic gain index (HGI) and heart hemodynamic reserve (HHR, **right panel**) and of established exercise-derived variables including circulatory power (CP), ventilatory efficiency (VE/VCO_2_ slope), percentage of predicted peak oxygen consumption (pVO_2_%), peak oxygen consumption per kilogram (pVO_2_/kg), and percentage of predicted peak heart rate (pHR%, **left panel**). Among conventional CPET indices, CP and VE/VCO_2_ slope demonstrated the highest discriminative ability (AUC 0.77 and 0.76), comparable to the HGI (AUC 0.74). When directly compared, HHR outperformed the HGI, showing superior predictive accuracy (AUC 0.78 vs. 0.74; *p* = 0.03). **Abbreviation**: CPET, cardiopulmonary exercise test.

**Table 1 diagnostics-15-02366-t001:** Baseline characteristics.

General Data	*n*: 479
Age (years)	59.7 (12)
Female, *n* (%)	76 (15.9)
BMI (kg/m^2^)	26.9 (4.2)
Ischemic etiology, *n* (%)	218 (45.5)
NYHA class, *n* (%)	
I	116 (24)
II	251 (52.5)
III	112 (23.5)
HF medications	
ACEi, *n* (%)	161 (34.9)
ARB, *n* (%)	33 (9.3)
ARNi, *n* (%)	250 (52.4)
Beta-blocker, *n* (%)	448 (93.9)
MRA, *n* (%)	381 (79.7)
SGLT2i, *n* (%)	113 (31.7)
Diuretics, *n* (%)	338 (71)
ICD, *n* (%)	186 (39.1)
CRT-D, *n* (%)	129 (27.1)
Laboratory variables	
Hb, g/dL	14.2 (13.1–15.3)
Na+, mEq/L	140 (139–142)
Creatinine, mg/dL	1.1 (0.9–1.3)
eGFR (MDRD), mL/min/m^2^	53.6 (42.2–67.8)
BNP, pg/mL	219 (97.5–478)
NT-proBNP, pg/mL	918 (362–1914)
Echographic variables	
LVEF, %	32.0 (7.9)
**HFrEF**, *n* (%)	393 (82)
**HFmrEF**, *n* (%)	86 (18)
sPAP, mmHg	34.4 (12.2)
Exercise variables	
Baseline SBP, mmHg	109.1 (15.1)
Baseline DBP, mmHg	69.5 (9.2)
Baseline HR, bpm	68.8 (12.4)
AT identified, *n* (%)	313 (87.7)
AT VO_2_, mL/min	954.4 (286.8)
Workload peak, Watts	100.4 (39.7)
RER	1.1 (0.1)
Peak SBP, mmHg	139.2 (26.9)
Peak DBP, mmHg	80.4 (12.2)
Peak HR, mmHg	115.9 (23.6)
pHR%	0.70 (0.1)
pVO_2_, mL/min	1347.7 (443.8)
pVO_2_/Kg, mL/min/Kg	16.9 (5.2)
pVO_2_%	60 (20)
EOV, *n* (%)	34 (9.6)
VE peak, L/min	57.3 (16.7)
RF peak, bpm	36.4 (12.5)
VE/VCO_2_ slope	34.9 (9.1)
CP, mmHg.mL/min/m^2^	2399.9 (992.2)
**HGI**	1.2 (0.7)
**HHR**	9.7 (9.5)

**Abbreviations**: ACEi, ACE inhibitors; AT, anaerobic threshold; BMI, body mass index; SBP, baseline systolic blood pressure; DBP, diastolic blood pressure; HR, baseline heart rate; CP, circulatory power; CRT, cardiac resynchronization therapy; CV, cardiovascular; EOV, exertional oscillatory breathing; HGI, hemodynamic gain index; HHR, heart hemodynamic reserve; HTX, heart transplantation; ICD, implantable cardioverter defibrillator; LVAD, left ventricular assist device; LVEF, left ventricular ejection fraction; NYHA, New York Heart Association; pHR%, peak heart rate expressed as percentage of maximum predicted; pVO_2_, peak oxygen consumption; RER, peak respiratory exchange ratio; RF, respiratory frequency; sPAP, systolic pulmonary artery pressure; VE/VCO_2_ slope, relationship between ventilation and carbon dioxide production.

**Table 2 diagnostics-15-02366-t002:** Cox univariate analysis for composite clinical outcome according to the main variables.

Variables	Hazard Ratio	95% CI	*p*-Value
Age (years)	1.01	0.98–1.03	0.573
Female	1.65	0.70–3.85	0.249
BMI (kg/m^2^)	0.98	0.92–1.04	0.495
Ischemic etiology	1.44	1.05–1.98	0.025
ACEi	1.42	0.82–2.44	0.209
ARB	0.59	0.18–1.95	0.387
ARNi	0.58	0.32–1.03	0.061
Beta-blocker	0.31	0.15–0.63	0.001
MRA	1.36	0.67–2.78	0.395
SGLT2i	4.26	1.80–10.1	<0.001
Diuretics	6.84	2.12–21.9	0.001
Hb, g/dL	0.84	0.72–0.98	0.026
LVEF, %	0.91	0.88–0.94	<0.001
sPAP, mmHg	1.03	1.01–1.05	0.002
pHR%	0.02	0.002–0.16	<0.001
pVO_2_/Kg, mL/min/Kg	0.89	0.85–0.94	<0.001
pVO_2_%	0.02	0.006–0.10	<0.001
EOV, *n* (%)	1.97	0.68–5.64	0.209
VE/VCO_2_ slope	1.06	1.04–1.08	<0.001
CP, mmHg.mL/min/m^2^	0.99	0.98–0.99	<0.001
**HGI**	0.22	0.12–0.39	**<0.001**
**HHR**	0.85	0.79–0.90	**<0.001**

Abbreviations: see Table 1.

**Table 3 diagnostics-15-02366-t003:** Cox multivariate model for composite clinical outcome: HGI and HHR treated as continuous variables.

Variables	Hazard Ratio	95% CI	*p*-Value
**HGI**	0.44	0.23–0.83	**0.011**
Beta-blocker	0.32	0.15–0.66	0.002
Diuretics	3.56	1.09–11.66	0.036
LVEF, %	0.95	0.91–0.98	0.006
Ischemic etiology	1.06	0.61–1.85	0.834
Hb, g/dL	1.00	0.86–1.17	0.967
pVO_2_%	0.20	0.02–1.93	0.164
VE/VCO_2_ slope	1.03	1.00–1.05	0.055
**Variables**	**Hazard Ratio**	**95% CI**	***p*-Value**
**HHR**	0.91	0.85–0.97	**0.005**
Beta-blocker	0.32	0.15–0.67	0.002
Diuretics	3.36	1.03–10.98	0.045
LVEF, %	0.95	0.91–0.99	0.008
Ischemic etiology	1.00	0.58–1.74	0.997
Hb, g/dL	1.01	0.87–1.19	0.860
pVO_2_%	0.21	0.021–2.14	0.189
VE/VCO_2_ slope	1.03	1.00–1.06	0.048

Abbreviations: see Table 1.

**Table 4 diagnostics-15-02366-t004:** Cox univariate and multivariate model for composite outcome: HGI and HHR treated as categorical variables.

HGI Tertile	Univariate Model			Multivariate Model		
	Hazard Ratio	95% CI	*p*	Adjusted Hazard Ratio	95% CI	*p*
Composite of CV death, LVAD implantation, and HTX						
1: <0.87 *n* = 143	12.77	3.94–41.45	<0.001	5.35	1.57–18.22	**0.007**
2: 0.87–1.43 *n* = 152	5.76	1.68–19.77	0.005	3.47	0.98–12.24	0.053
3: >1.43 *n* = 148	1.00	-	-	1.00	-	-
**HHR Tertile**	**Univariate Model**			**Multivariate Model**		
	**Hazard Ratio**	**95% CI**	** *p* **	**Adjusted Hazard Ratio**	**95% CI**	** *p* **
Composite of CV death, LVAD implantation, and HTX						
1: <4 *n* = 151	9.03	3.56–22.9	<0.001	3.75	1.37–10.25	**0.010**
2: 4–10 *n* = 142	2.51	0.87–7.24	0.08	1.60	0.54–4.73	0.397
3: >10 *n* = 150	1.00	-	-	1.00	-	-

**Multivariable model**: BB treatment, diuretic therapy, LVEF%, ischemic etiology, hemoglobin, pVO_2_%, and VE/VCO_2_ slope. **Abbreviations**: see Table 1.

## Data Availability

The data presented in this study are available on request from the corresponding author. The data are not publicly available due to privacy and ethical restrictions.

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
