# Peer review of "Clinical and Prognostic Impact of Hemodynamic Gain Index and Heart Hemodynamic Reserve in Heart Failure with Reduced and Mildly Reduced Ejection Fraction: A Multicenter Study"

_diagnostics, 2025, doi:10.3390/diagnostics15182366_

Round 1
Reviewer 1 Report
Comments and Suggestions for Authors
This article was well written, however some issues need to be explained.
- Abstract is ok.
- Method section of the main text
- I know whether the patients valve disease were excluded from the presented study or not?
-
How did you determine the parameters to be included in the multivariate analysis?
Results
1. I wonder that is the prognostic Impact of Hemodynamic Gain Index
and Heart Hemodynamic Reserve's effect similar in the reduced and slightly reduced EF groups?
Discussion
1.In the first sentence of your discussion, you should state what you found in this study.
Author Response
For research article
|
Response to Reviewer X Comments
|
||
|
1. Summary |
|
|
|
Thank you for taking the time to review this manuscript. Please find the detailed responses below and the corresponding revisions highlighted in the re-submitted files.
|
||
|
2. Point-by-point response to Comments and Suggestions for Authors |
||
|
Comment 1: I know whether the patients valve disease were excluded from the presented study or not? |
||
|
Response 1: Thank you for pointing this out. We agree with your comment. Patients with severe primary mitral or aortic valve disease were excluded from this analysis. We have revised the Study Population section accordingly (page 2, lines 88–89). |
||
|
Comment 2: How did you determine the parameters to be included in the multivariate analysis? |
||
|
Response 2: We thank the Reviewer for this appropriate question and the opportunity to clarify. After conducting univariate analyses, we selected variables that were statistically associated with the outcome for inclusion in the multivariate model. The number of variables included respected the rule of 10 events per variable, in order to preserve the reliability of the results. Variables related to exercise were preferentially selected, to strengthen the message that HGI and HHR maintain their prognostic relevance even after adjustment for other established CPET metrics (see page 3, lines 137–141). |
||
|
Comment 3: I wonder that is the prognostic Impact of Hemodynamic Gain Index
Response 3: We thank the Reviewer for raising this interesting point, which was also highlighted by other reviewers. Initially, we conducted this analysis and included LVEF dichotomized at 50% as a subgroup variable in the Forest plot. However, since only one event occurred in the HFmrEF group, the resulting confidence interval was extremely wide and the estimate was statistically unreliable. For this reason, we did not report it as a main result, and instead explained this limitation in the Study Limitations section (see page 13, lines 362-363).
Comment 4: In the first sentence of your discussion, you should state what you found in this study.
Respone 4: We thank the Reviewer for this valuable suggestion. In accordance with this and other Reviewers’ requests, we have revised the Discussion section by introducing an incipit that briefly summarizes the main results of our study (see page 11, lines 271–276).
|
||
|
|
||
|
|
||
Reviewer 2 Report
Comments and Suggestions for Authors
The Editor, Sept 3,2025, diagnostics Diagnostics Clinical and Prognostic Impact of Hemodynamic Gain Index and Heart Hemodynamic Reserve in Heart Failure with Reduced and Mildly Reduced Ejection Fraction. A Multicentre Study. by Emiliano Fiori et al. 1.Abstract. Need English editing. 2.Subjects and Methods is better than Methods 3.Recruitment of patients. 4. Criteria of diagnosis. 5.Inclusion criteria. 6.Exclusion criteria. 7.Collection of Clinical data 8.Echocardiography. 9.Cardiopulmonary Exercise test. Please divide in 2- paras. 10.Why do you write: Study Objectives again, under the section of Methods, because you already did it under the section of Introduction. If it is necessary to explain statistics, please give it under this section, but there is no need to make it a subheading. 11. Statistical analysis. Take opinion from experts. 12.Results. Table 1, should include data for both genders separately, then total may also given. The clinical features and complications may differ among men and women, because of the variotions in sex as well as exposome. Avoid short forms in the table unless necessary, because its inconvenient for the audience. Data Men Women Total 13.Discussion Write your most important finding in the first sentence then discuss other studies from yur country and elsewhere. Conclusions should be summarized. 14.References Give 1-2 references from 2025

improve
Author Response
For research article
|
Response to Reviewer X Comments
|
||
|
1. Summary |
|
|
|
Thank you for taking the time to review this manuscript. Please find the detailed responses below and the corresponding revisions highlighted in the re-submitted files.
|
||
|
2. Point-by-point response to Comments and Suggestions for Authors |
||
|
Comment 2-8: Subjects and Methods is better than Methods. Recruitment of patients. Criteria of diagnosis. Inclusion criteria. Exclusion criteria. Collection of Clinical data. Echocardiography. |
||
|
Response 2-8: We thank the Reviewer for this helpful suggestion. To improve readability, we have restructured the Methods section. The recruitment and data acquisition process is now presented in a more organized and fluid manner. |
||
|
Comment 9: Cardiopulmonary Exercise test. Please divide in 2- paras. |
||
|
Response 9: We thank the Reviewer for this suggestion. Although the paragraph is relatively long, it provides a unified description of how the key CPET variables under investigation were measured or calculated. Dividing it into two paragraphs (for instance, separating the HGI and HHR descriptions) would be difficult, as other variables are described in the first part and such a distinction would be redundant. For this reason, we have retained the current structure.
Comment 10: Why do you write: Study Objectives again, under the section of Methods, because you already did it under the section of Introduction. If it is necessary to explain statistics, please give it under this section, but there is no need to make it a subheading. Response 10: We thank the Reviewer for this observation. We believe that briefly and precisely restating the study objectives within the Methods section improves clarity in linking the aims with the subsequent statistical analysis, and therefore we have retained this subsection.
|
||
|
Comment 11: Statistical analysis. Take opinion from experts. Response 11: We thank the Reviewer for this comment. We would like to clarify that the statistical analyses for this manuscript were performed by a professional medical statistician with extensive experience in the field. We are confident in the robustness of the methods employed; however, we would be happy to address any specific concerns or suggestions regarding the analyses, should the Reviewer consider them necessary.
Comment 12: Table 1, should include data for both genders separately, then total may also given. The clinical features and complications may differ among men and women, because of the variotions in sex as well as exposome. Avoid short forms in the table unless necessary, because its inconvenient for the audience. Response 12: We thank the Reviewer for this thoughtful suggestion. We agree that gender-specific analyses may provide important insights; however, this goes beyond the scope of the present manuscript. Moreover, as acknowledged in the Study Limitations section, the underrepresentation of women in our cohort would substantially limit the statistical reliability of such an analysis. With regard to abbreviations, we used internationally recognized short forms in the tables and provided comprehensive legends to ensure clarity for the reader (see Table 1).
Comment 13: Write your most important finding in the first sentence then discuss other studies from yur country and elsewhere. Conclusions should be summarized. Response 13: We thank the Reviewer for this valuable suggestion. In accordance with this and other Reviewers’ requests, we have revised the Discussion section by introducing an incipit that briefly summarizes the main results of our study (see page 11, lines 271–276). Conclusions have been shortened and made more concise (see page 13, lines 375-382).
Comment 14: Give 1-2 references from 2025. Response 14: We thank the Reviewer for this suggestion. We have updated the reference list concerning a recent review article (see Ref. 13), which was initially available online in 2024 and formally published in 2025.
4. Response to Comments on the Quality of English Language |
||
|
Point 1: improve. Abstract: Need English editing |
||
|
Response 1: We thank the Reviewer for this remark. A native English speaker was involved in the revision process and assisted us in improving the language throughout the manuscript. . |
||
|
|
||
Reviewer 3 Report
Comments and Suggestions for Authors
This multicenter retrospective cohort study explored the prognostic value of hemodynamic gain index and heart hemodynamic reserve in patients with HFrEF and HFmrEF, particularly compared with the broadly adopted metabolic parameters in CPET. The authors concluded that both HGI and HHR are non-metabolic, easy-to-measure, and reliable prognostic markers in HF patients with EF < 50%. The study was well designed, and the methods were appropriately performed. The results were discussed, and the limitations were listed.
Some major concerns:
- In “3.2 Functional characterization of HGI and HHR” in the results, the specific data of the variables should be presented, and statistical analysis, such as one-way ANOVA and post-hoc analysis, should be conducted to evaluate the significant difference and confirm the trend.
- In Table 2 and Table 3, it is better to show the P values rather than merely “NS”, because the value could tell how close the result was to significance.
- In the multivariable Cox regression model analysis, the variables that were adjusted for were not sufficient. More key confounding factors, such as age, BMI, and hypertension, should be included.
- Could you involve HFmrEF and HFrEF in the subgroup analysis to further elucidate the prognostic value of HGI and HHR in terms of different EF?
Author Response
For research article
|
Response to Reviewer X Comments
|
||
|
1. Summary |
|
|
|
Thank you for taking the time to review this manuscript. Please find the detailed responses below and the corresponding revisions highlighted in the re-submitted files.
|
||
|
2. Point-by-point response to Comments and Suggestions for Authors |
||
|
Comment 1: In “3.2 Functional characterization of HGI and HHR” in the results, the specific data of the variables should be presented, and statistical analysis, such as one-way ANOVA and post-hoc analysis, should be conducted to evaluate the significant difference and confirm the trend. Response 1: We thank the Reviewer for this valuable comment, which highlighted an important gap that needed to be addressed. Accordingly, we compared CPET variables across HGI and HHR tertiles using the Kruskal–Wallis test. All post-hoc pairwise comparisons between tertiles were performed with the Dunn test and Bonferroni correction for multiple testing, and all reached statistical significance (P < 0.05). All four CPET variables showed a clear worsening trend from T3 to T1. To preserve readability of the main text, we preferred not to add another table in the body of the manuscript; instead, we have included the results in the Supplementary Materials (see page 6, line 181).
Comment 2: In Table 2 and Table 3, it is better to show the P values rather than merely “NS”, because the value could tell how close the result was to significance. |
||
|
Response 2: Thank you for pointing this out. We agree with this comment. Therefore, we have provided all the P values in both tables. |
||
|
Comment 3: In the multivariable Cox regression model analysis, the variables that were adjusted for were not sufficient. More key confounding factors, such as age, BMI, and hypertension, should be included. |
||
|
Response 3: We thank the Reviewer for this important comment. We fully agree that potential confounding factors such as age, BMI, and hypertension are clinically relevant. However, as the number of events in our cohort was limited, including a larger set of covariates would have compromised the stability and reliability of the multivariable Cox regression model. To preserve statistical validity, we adhered to the conventional rule of at least 10 events per variable. Within this constraint, we prioritized variables that were statistically associated with outcomes in univariate analyses, with particular attention to exercise-related parameters, in order to evaluate whether HGI and HHR retained independent prognostic value beyond established CPET metrics (see page 3, lines 137–141). We acknowledge this as a limitation of our analysis and have clarified it in the Study Limitations section (see page 13, lines 362-363).
Comment 4: Could you involve HFmrEF and HFrEF in the subgroup analysis to further elucidate the prognostic value of HGI and HHR in terms of different EF? Response 4: We thank the Reviewer for raising this interesting point, which was also highlighted by other reviewers. Initially, we conducted this analysis and included LVEF dichotomized at 50% as a subgroup variable in the Forest plot. However, since only one event occurred in the HFmrEF group, the resulting confidence interval was extremely wide and the estimate was statistically unreliable. For this reason, we did not report it as a main result, and instead explained this limitation in the Study Limitations section (see page 13, lines 362-363). |
||
|
|
||
Round 2
Reviewer 1 Report
Comments and Suggestions for Authors
All revisions are ok.
Author Response
We thank the Reviewer for his/her helpful comments, which have improved our manuscript.
Reviewer 3 Report
Comments and Suggestions for Authors
The authors have addressed my concerns and given the satisfactory responses. I have no further comments on the revised manuscript.
Author Response

(The authors gave the same response as above.)
